# Targeted NanoBiT Screening Identifies a Novel Interaction Between SNAPIN and Influenza A Virus M1 Protein

**DOI:** 10.3390/biology14121770

**Published:** 2025-12-11

**Authors:** Xiaoxuan Zhang, Huanhuan Wang, Conghui Zhao, Wenjun Shi, Faxin Wen, Haoxi Qiang, Sha Liu, Peilin Li, Xinhui Chen, Chunping Zhang, Jiacheng Huang, Yang Wang, Ziyi Zhang, Shujie Ma

**Affiliations:** 1Joint Laboratory of Animal Pathogen Prevention and Control of Fujian-Nepal, College of Animal Sciences, Fujian Agriculture and Forestry University, Fuzhou 350002, China; zhangxiaoxuan5426@163.com (X.Z.); wanghuan12282487@163.com (H.W.); zhaoconghui19@163.com (C.Z.); 13592554517@163.com (F.W.); siriusxx_010106@163.com (H.Q.); m15037362321@163.com (S.L.); lpl0051129@163.com (P.L.); 18698331895@163.com (X.C.); m18806027576@163.com (C.Z.); m13395087570@163.com (J.H.); wangyangvv2023@163.com (Y.W.); 17350500508@163.com (Z.Z.); 2Technology Center of Qingdao Customs, Qingdao 266000, China; duoyanji@126.com; 3Key Laboratory of Animal Pathogen Infection and Immunology of Fujian Province, College of Animal Sciences, Fujian Agriculture and Forestry University, Fuzhou 350002, China; 4Key Laboratory of Fujian-Taiwan Animal Pathogen Biology, College of Animal Sciences, Fujian Agriculture and Forestry University, Fuzhou 350002, China

**Keywords:** Influenza A virus, NanoBiT, protein-protein interaction, SNAPIN, M1

## Abstract

Influenza A virus (IAV) evolves quickly, making it a continued threat to human health. To better understand how the virus interacts with host cells, we developed a new live-cell screening method using the NanoBiT system to rapidly detect protein–protein interactions (PPIs) in living cells. Using this approach, we discovered potential novel interactions between the host protein SNAPIN and three influenza virus-encoded proteins: M1, M2, and NS2. To confirm the platform’s reliability, we selected the SNAPIN-M1 interaction for further validation using two additional established laboratory techniques. This study provides a reliable and efficient screening platform and lays the foundation for future research into how SNAPIN may influence influenza virus replication.

## 1. Introduction

Influenza A virus (IAV) poses a significant zoonotic threat, continually endangering both animal and human health. Specifically, H1N1 and H3N2 subtypes have been adapted to sustain human-to-human transmission, causing recurrent epidemics and pandemics that threaten human populations worldwide [1]. Avian-origin subtypes, in particular H5 and H7 highly pathogenic avian influenza viruses (HPAIVs), cause substantial economic losses in the global poultry industry, with certain strains of these two subtypes demonstrating zoonotic transmission to humans [2,3,4,5,6].

IAV is an enveloped virus with a segmented, single-stranded, negative-sense RNA genome, belonging to the member of the *Orthomyxoviridae* family. The genome of IAV encodes at least 18 proteins, including ten basic proteins and eight accessory proteins [7]. The ten basic proteins include polymerase basic protein 2 (PB2), polymerase basic protein 1 (PB1), polymerase acidic protein (PA), hemagglutinin (HA), nucleoprotein (NP), neuraminidase (NA), matrix protein 1 (M1), matrix protein 2 (M2), nonstructural protein 1 (NS1), and nonstructural protein 2 (NS2, also referred to as nuclear export protein, NEP) [8]. Many host proteins are involved in regulating IAV replication by directly interacting with these virus-encoded basic proteins. Consequently, understanding the intricate host–virus protein–protein interaction (PPI) network is key to elucidating viral infection mechanisms and pathogenesis. Multiple host restriction factors, such as transport protein particle complex 6A (TRAPPC6A) [9], phospholipid scramblase 1 (PLSCR1) [10], free fatty acid receptor 2 (FFAR2) [11], inhibit IAV replication at various infection stages through such interactions. Conversely, IAVs hijack host proteins like metabotropic glutamate receptor subtype 2 (mGluR2) [12], immunoglobulin superfamily DCC subclass member 4 (IGDCC4) [13], and Bcl10-interacting protein with CARD (BinCARD) [14] to facilitate replication and enhance infectivity. Critically, studies defining these PPIs have proven essential for revealing the functional roles of host proteins during IAV replication. Thus, establishing rapid screening methods for host–virus PPIs is vital for advancing mechanistic studies of viral pathogenesis and host adaptation.

Our previous findings have shown that there is a decrease in SNAPIN levels during IAV infection both in vitro and in vivo, suggesting a potential role for SNAPIN in the viral replication cycle [15]. In this study, we employed Nanobit technology to screen for virus-encoded proteins that interact with SNAPIN and subsequently validated the specific interaction between SNAPIN and M1 protein using co-immunoprecipitation (Co-IP) and glutathione S-transferase (GST) pull-down assays. Collectively, we successfully established a live-cell screening platform based on NanoBiT system for rapid PPI discovery between host and virus-encoded proteins.

## 2. Materials and Methods

### 2.1. Cells, Eggs, and Viruses

Human lung adenocarcinoma epithelial cells (A549, CCL-185) and human embryonic kidney cells (HEK293T, CRL-3216) were obtained from the American Type Culture Collection (ATCC, Manassas, VA, USA). The A549 cell line stably overexpressing Flag-tagged SNAPIN (designated as A549-Flag-SNAPIN) was established using lentiviral transduction. The human SNAPIN coding sequence (NM_012437.6) was cloned into a pLVX-IRES-puro vector with an N-terminal 3 × Flag tag using FastDigest BamHI (FD0054, Thermo Fisher Scientific, Waltham, MA, USA) and FastDigest EcoRI (FD0275, Thermo Fisher Scientific, Waltham, MA, USA). The recombinant plasmid was then packaged into lentiviruses in HEK293T cells. Subsequently, A549 cells were infected with the lentiviruses and selected with puromycin (CNSBR008, Sigma-Aldrich, Darmstadt, Germany) to generate a stable Flag-SNAPIN expression cell line. All cells were maintained in Dulbecco’s modified Eagle’s medium (DMEM) (C11995500BT, Thermo Fisher Scientific, Waltham, MA, USA) supplemented with 10% fetal bovine serum and antibiotics at 37 °C in a humidified 5% CO_2_ atmosphere. Specific pathogen-free (SPF) embryonated chicken eggs were purchased from Jinan SPAFAS Poultry Co., Ltd. (Jinan, China). Virus stocks of A/WSN/33 (H1N1) (abbreviated as WSN) were propagated in 9-day-old SPF embryonated chicken eggs and stored at −70 °C before use.

### 2.2. Construction of Plasmids

The ten gene segments (PB2, PB1, PA, HA, NP, NA, M1, M2, NS1, and NS2) of WSN were individually cloned into the LgBiT vector (N2014, Promega, Madison, WI, USA) using the ClonExpress II One Step Cloning Kit (C112, Vazyme, Nanjing, China). The recombinant plasmids were named as LgN-PB2, LgN-PB1, LgN-PA, LgN-HA, LgN-NP, LgN-NA, LgN-M1, LgN-M2, LgN-NS1, and LgN-NS2, respectively. Similarly, the human SNAPIN coding sequence (NM_012437.6) was amplified from cDNA derived from A549 cells and inserted into the SmBiT vector (N2014, Promega, Madison, WI, USA) using the ClonExpress II One Step Cloning Kit (C112, Vazyme, Nanjing, China), generating the construct designated as SmN-SNAPIN. The Myc-SNAPIN and Flag-M1 plasmids were constructed by inserting the human SNAPIN coding sequence (NM_012437.6) and the WSN M1 segment into a pCAGGS vector with an N-terminal Myc or Flag tag, respectively, using the ClonExpress II One Step Cloning Kit (C112, Vazyme, Nanjing, China). All plasmids were verified by DNA sequencing.

### 2.3. Antibodies

The following antibodies were purchased from commercial sources: rabbit anti-Flag polyclonal antibody (pAb) (20543-1-AP, Proteintech, IL, USA), rabbit anti-Myc pAb (16286-1-AP, Proteintech, IL, USA), mouse anti-M1 pAb (GTX125928, GeneTex, Irvine, CA, USA), rabbit anti-M2 mAb (GTX637581, GeneTex, Irvine, CA, USA) mouse anti-GAPDH mAb (60004-1-Ig, Proteintech, IL, USA), mouse anti-β actin mAb (66009-1-Ig, Proteintech, IL, USA), anti-LgBiT monoclonal antibody (N7100, Promega, Madison, WI, USA), Alexa Fluor 488–conjugated goat anti-mouse IgG (A11008, Thermo Fisher Scientific, Waltham, MA, USA). For Western blotting, the following secondary antibodies were used: Peroxidase-conjugated AffiniPure Goat Anti-Mouse IgG (H + L) (115-035-003, Jackson ImmunoResearch, West Grove, PA, USA) and Peroxidase-conjugated AffiniPure Goat Anti-Rabbit IgG (H + L) (111-035-003, Jackson ImmunoResearch, West Grove, PA, USA). Specific proteins were visualized using an enhanced chemiluminescence (ECL) detection system (P10100, NCM Biotech, Suzhou, China).

### 2.4. Confocal Microscopy

A549 cells were seeded on glass-bottom dishes and transfected with the indicated plasmids. At 26 h post-transfection (hpt), cells were fixed with 4% paraformaldehyde in PBS for 15 min and permeabilized with 0.5% Triton X-100 (ST1723, Beyotime, Shanghai, China) in PBS for 30 min. After blocking with 5% BSA in PBS for 1 h, cells were incubated with anti-LgBiT monoclonal antibody (1:200) at 4 °C overnight, followed by three PBS washes and incubation with Alexa Fluor 488–conjugated goat anti-mouse IgG for 1 h. Nuclei were stained with DAPI (C1002, Beyotime, Shanghai, China) for 15 min. Images were acquired using a Leica laser scanning confocal microscope (STELLARIS5, Leica, Wetzlar, Germany).

### 2.5. NanoBiT Luciferase Complementation Experiment

HEK293T cells were seeded in a white opaque 96-well plate and allowed to reach 80% confluence. Then, each well was co-transfected with 200 ng of the SmN-SNAPIN plasmid along with one of the ten virus-encoded protein expression plasmids (LgN-PB2, LgN-PB1, LgN-PA, LgN-HA, LgN-NP, LgN-NA, LgN-M1, LgN-M2, LgN-NS1, or LgN-NS2), with each viral protein expression plasmid also at 200 ng, using Lipo8000 transfection reagent (C0533, Beyotime, Shanghai, China) according to the manufacturer’s instructions. In parallel, positive control pair plasmids (LgBiT-PRKAR2A and SmBiT-PRKACA) as well as the negative control pair plasmids from the NanoBiT kit were transfected separately, with each control plasmid also transfected at 200 ng per well. After 24 h, the medium was replaced with 100 µL of fresh DMEM per well. Prior to measurement, 25 µL of diluted substrate from the Nano-Glo Live Cell Assay System (N2011, Promega, Madison, WI, USA) was added to each well, and luminescence was measured using a GloMax 96 microplate luminometer (GM2010, Promega, Madison, WI, USA) from Promega.

### 2.6. Protein Interaction Prediction

As IAV M1 protein is known to form dimer or oligomer, the SNAPIN-M1 interaction prediction was conducted using a dimeric M1 structure and the full-length human SNAPIN protein [16,17]. Structural prediction of the potential interacting domain between human SNAPIN and IAV M1 protein was performed using AlphaFold 3 [18]. The full-length sequences of SNAPIN and M1 were submitted to the AlphaFold 3 online server (https://alphafoldserver.com/, accessed on 1 September 2025), with the M1 copy number set to two and all other parameters kept at default settings. The predicted complex structure was subsequently visualized and analyzed using UCSF Chimera X (version 1.9) [19]. 

### 2.7. Co-Immunoprecipitation Experiment (Co-IP)

HEK293T cells were transfected with the indicated plasmids using Lipo8000 transfection reagent (C0533, Beyotime, Shanghai, China). After 36 hpt, cells were lysed in immunoprecipitation lysis buffer (P0013, Beyotime, Shanghai, China) supplemented with 1 mM phenylmethylsulfonyl fluoride (PMSF) (ST505, Beyotime, Shanghai, China) for 30 min on ice. The lysates were then centrifuged at 12,000 rpm for 10 min at 4 °C. For the Co-IP assay, 10% of the cleared lysate supernatant was reserved as the input control, and the remaining 90% was incubated with the indicated antibody-conjugated magnetic beads. The supernatants were incubated with either anti-Flag VHH magnetic beads (SB-NM018, ShareBio, Shanghai, China) or anti-Myc VHH magnetic beads (SB-NM019, ShareBio, Shanghai, China) for 1–3 h at 4 °C with gentle rocking. Following incubation, the beads were washed four times with ice-cold PBS containing 1 mM phenylmethanesulfonyl fluoride (PMSF) (ST506, Beyotime, Shanghai, China). The bound proteins were then boiled in 2 × SDS loading buffer, separated by 10% SDS-PAGE, and analyzed by Western blotting.

### 2.8. Cell Infection

A549 cells cultured in 6 cm dishes were infected with WSN at a multiplicity of infection (MOI) of 1 for 1 h. After infection, the cells were washed three times with PBS and then maintained in DMEM supplemented with 0.125 mg/mL tosylsulfonyl phenylalanyl chloromethyl ketone (TPCK)-treated trypsin at 37 °C. At 24 h post-infection (hpi), the cells were harvested for Co-IP assays. A549-Flag-SNAPIN cells were grown on 12-well plates and infected with WSN virus at an MOI of 1. The inoculum was removed after 1 h of incubation. After three washes with PBS, the cells were supplemented with DMEM containing 0.125 mg/mL TPCK-treated trypsin and were incubated at 37 °C. Virus-containing culture supernatant was collected at 24 hpi and titrated in MDCK cells. NP protein was tested by Western blotting.

### 2.9. GST Pull-Down

GST-tagged SNAPIN was expressed in *E. coli* BL21(DE3) and purified using a Mag-Beads GST Fusion Protein Purification Kit (C650031, Sangon Biotech, Shanghai, China) following the manufacturer’s instructions. HEK293T cells grown in 6 cm dishes were transfected with 5 μg of either empty Flag-vector or Flag-M1 plasmid using Lipo8000 transfection reagent (C0533, Beyotime, Shanghai, China). At 48 hpt, the cells were lysed in IP buffer containing 1 mM PMSF for 30 min on ice, followed by centrifugation at 12,000 rpm for 10 min at 4 °C. The supernatants were incubated with anti-Flag magnetic beads (HY-K0207, MedChemExpress, Monmouth Junction, NJ, USA) overnight at 4 °C with gentle agitation and then washed three times with ice-cold PBS containing 1 mM PMSF. Ten percent of the clarified lysate was saved as the input, and the remaining fraction was incubated with Mag-Beads pre-loaded with equal masses of purified GST or GST-SNAPIN. Purified Flag-tagged M1 protein was mixed with Mag-Beads bound to either GST or GST-SNAPIN. Equal masses of GST or GST-SNAPIN were loaded onto Mag-Beads, ensuring equivalent bait input across all reactions. The mixtures were rocked for 30 min at room temperature, washed three times with ice-cold PBS, separated by SDS-PAGE, and finally stained with BeyoBlue Plus Coomassie Blue Super Fast Staining Solution (P0003S, Beyotime, Shanghai, China).

## 3. Results

### 3.1. NanoBiT Screening of SNAPIN–IAV Protein Interactions

NanoBiT system was employed to establish quantitative analysis of PPIs within living cells under physiological conditions. Specifically, protein A and protein B were fused to LgBiT and SmBiT subunits, designated as LgBiT-A and SmBiT-B, respectively. Upon co-expression, the interaction between LgBiT-A and SmBiT-B brings the fused NanoBiT subunits into proximity, reconstituting functional luciferase that emits a bright luminescent signal (Figure 1A). Therefore, detection of the luminescent signal enables quantitative determination of the PPIs.

Our previous study showed that SNAPIN is downregulated during IAV infection, indicating its potential regulatory role in the viral lifecycle [15]. To screen for interactions between SNAPIN and virus-encoded proteins, the LgBiT fragment was fused to the N-terminus of the ten basic viral proteins, generating constructs designated as LgN-PB2, LgN-PB1, LgN-PA, LgN-HA, LgN-NP, LgN-NA, LgN-M1, LgN-M2, LgN-NS1, and LgN-NS2, respectively. Conversely, the SmBiT fragment was fused to the N-terminus of SNAPIN protein, creating the construct SmN-SNAPIN. LgBiT-PRKAR2A co-transfected with SmBiT-PRKACA served as the positive control for interaction, while the LgN-tagged virus-encoded proteins co-transfected with SmN-HaloTag acted as the negative control pairs. All plasmids were validated by double-restriction digestion (Appendix A) and DNA sequencing. To verify expression from the constructed plasmids, we performed confocal laser microscopy and observed that all corresponding proteins were successfully expressed (Appendix A). HEK293T cells were co-transfected with 200 ng of SmN-SNAPIN plasmid and individual LgN-tagged virus-encoded protein expression constructs. Parallel control transfections included the positive control (LgBiT-PRKAR2A/SmBiT-PRKACA) and negative control pair plasmids. Luminescence intensity was quantified 24 hpt using a GloMax-96 microplate luminometer according to manufacturer’s protocols. Luminescence assays revealed a 354.62-fold signal increase for the positive control relative to the negative control. Among the ten virus-encoded proteins tested, significant signals were detected only for SNAPIN’s interactions with M1, M2, and NS2 with respective increases of 5.12-fold, 8.62-fold, and 2.37-fold, respectively. Based on the manufacturer’s guidelines, the results suggest the presence of specific, albeit weaker, potential interactions between SNAPIN and the M1, M2, and NS2 proteins (Figure 1B). Collectively, these NanoBiT luciferase complementation assays demonstrate that M1, M2, and NS2 may specifically interact with SNAPIN.

### 3.2. Structural Prediction of PPI

To gain preliminary insights into the potential binding interface between SNAPIN and the IAV M1 protein, we performed a computational structural prediction using AlphaFold 3. As the M1 protein is known to function as a dimer, the prediction was conducted using a dimeric full-length M1 structure and the full-length human SNAPIN protein. The predicted complex structure was visualized using UCSF ChimeraX (version 1.9). In the model, the M1 dimer is represented as a molecular surface, with the two monomers colored in light purple and light magenta, respectively. The SNAPIN protein is depicted as a cartoon, with its N-terminal region colored yellow and its C-terminal region colored orange (Appendix A). This analysis suggested that the N-terminal domain of SNAPIN may engage more closely with the dimer interface of the M1 protein compared to its C-terminal domain.

### 3.3. Co-IP Assays

To biochemically validate the NanoBiT PPI results, we focused on the SNAPIN-M1 interaction using Co-IP assays. HEK293T cells were transfected with Myc-tagged SNAPIN and Flag-tagged M1 expression constructs, either individually or in combination. At 24 hpt, cell lysates were immunoprecipitated with anti-Flag VHH magnetic beads, followed by Western blotting analysis. Notably, Flag-tagged M1 was specifically co-precipitated with Myc-tagged SNAPIN only upon co-expression, indicating an interaction of SNAPIN and M1 (Figure 2A). To establish reciprocity, reverse Co-IP was performed using anti-Myc VHH magnetic beads for immunoprecipitation. Consistent with the initial findings, Myc-tagged SNAPIN was again co-precipitated with Flag-tagged M1 (Figure 2B), further demonstrating the specificity of this interaction. To investigate SNAPIN and virus-encoded protein interactions in the context of viral infection, we infected A549-Flag-SNAPIN cells with WSN. Cell lysates collected at 24 hpi were subjected to Co-IP with anti-Flag VHH magnetic beads. Critically, endogenous M1 and M2 proteins were both co-immunoprecipitated with Flag-SNAPIN in infected A549-Flag-SNAPIN cells (Figure 2C), indicating that the SNAPIN-M1 and SNAPIN-M2 interaction occurs during active viral replication. To further explore the functional relevance of these interactions, we examined the effect of SNAPIN overexpression on viral replication using an A549-Flag-SNAPIN cells. We found that SNAPIN overexpression suppressed WSN virus replication at 24 hpt (Appendix A), suggesting that SNAPIN plays an important role during IAV replication, potentially through its interactions with the M1 and M2 proteins. Collectively, these complementary Co-IP results, together with the functional data, validate the reliability of the NanoBiT screening outcomes.

### 3.4. Detection of SNAPIN–M1 Interaction by GST Pull-Down

To determine whether SNAPIN and M1 interact directly, we performed GST pull-down assays. Recombinant GST and GST-SNAPIN fusion proteins were expressed in *E. coli* BL21 (DE3) and purified using a Mag-Beads GST Fusion Protein Purification Kit. In parallel, HEK293T cells cultured in 6 cm dishes were transfected with 5 μg of either empty Flag-vector or Flag-M1 expression plasmid using Lipo8000 reagent (C0533, Beyotime, Shanghai, China). At 48 hpt, lysates from Flag-M1 expressing cells were incubated with purified GST or GST-SNAPIN proteins. GST, GST-SNAPIN, and Flag-M1 were detected by Western blotting in lysate samples (Figure 3A). Following affinity capture and extensive washing, bound protein complexes were separated by SDS-PAGE. GST and GST-SNAPIN were detected by Coomassie blue (CB) staining (Figure 3B). M1 protein was pulled down by GST-SNAPIN, but not by GST alone (Figure 3B). These results indicate that SNAPIN directly interacts with M1 protein in vitro, independent of cellular cofactors.

## 4. Discussion

NanoBiT system has been used in many studies including drug screening, signal transduction analysis, intracellular localization imaging, PPIs, and viral infection mechanism analysis [20,21,22]. In this study, we established a rapid and reliable method for screening interactions between host proteins and viral proteins using NanoBiT technology (Promega, Madison, WI, USA), demonstrated through the identification of SNAPIN-viral protein interactions.

Traditional methods for studying PPIs include the yeast two-hybrid system (Y2H), bacterial two-hybrid system (BACTH), GST pull-down, tandem affinity purification (TAP), fluorescence resonance energy transfer (FRET), bimolecular fluorescence complementation (BiFC), and co-immunoprecipitation (Co-IP). However, each of these established techniques has inherent limitations. For instance, Y2H, BACTH, and GST pull-down are typically performed in heterologous systems (yeast or bacteria) and may lack the specific post-translational modifications found in native eukaryotic cellular environments [9,23,24]. FRET suffers from low signal-to-noise ratios and complex instrumentation requirements, while BiFC is hindered by irreversible complementation and false-positive artifacts from spontaneous fragment assembly [25]. Co-IP experiments, while powerful for detecting stable complexes, are often less effective at capturing transient or weak protein interactions and frequently require combination with other methods for validation or enhanced detection. To overcome challenges like detecting transient interactions sensitively in physiological environments, the NanoBiT system offers a powerful alternative.

The NanoBiT system utilizes engineered fragments of NanoLuc luciferase: a large subunit (LgBiT, 17.6 kDa) and a small peptide tag (SmBiT, 1.3 kDa). These subunits are expressed stably in isolation but remain enzymatically inactive. However, when brought into close proximity by interacting proteins fused to them, they spontaneously reassemble to form functional NanoLuc luciferase, producing intense bioluminescence [26,27]. Owing to its exceptional sensitivity for detecting weak or transient interactions, remarkable stability, and relative experimental simplicity compared to traditional methods, NanoBiT has rapidly gained widespread adoption. It is now extensively applied across diverse research areas, including real-time monitoring of protein degradation kinetics, high-throughput screening for pathogen virulence factors, and detailed characterization of PPIs such as dimerization dynamics [27,28,29,30].

SNAPIN was first discovered for its role in neurotransmission, where it functions by binding to SNAP25 within the SNARE complex to mediate vesicle fusion [31]. More recent studies have expanded its functional profile, demonstrating that SNAPIN is vital for maintaining neuronal health by coordinating late endosome-lysosome transport, lysosome maturation, and autophagic processes [32,33]. SNAPIN frequently engages in direct interactions with specific viral proteins to regulate viral replication. For instance, the porcine reproductive and respiratory syndrome virus (PRRSV) exploits SNAPIN through its GP5 and M proteins to facilitate intracellular transport and membrane fusion [34]. Similarly, multiple human cytomegalovirus (HCMV) proteins, including PUL130, UL142, and UL70, bind SNAPIN to modulate viral replication [35,36,37]. Our previous study revealed that SNAPIN is significantly downregulated during IAV infection, indicating its regulatory role in the viral lifecycle [15]. Consistent with this, overexpression of SNAPIN in A549 cells followed by WSN infection showed that SNAPIN suppresses viral replication (Appendix A). These findings prompted us to hypothesize that SNAPIN may also interact with IAV-encoded proteins. Given its established function in autophagy and lysosomal protein degradation, we propose that SNAPIN may engage viral proteins and facilitate their degradation, thereby restricting viral replication. To investigate this possibility, we used a NanoBiT-based screening system to examine interactions between SNAPIN and ten basic IAV proteins. This rapid approach identified previously unreported interactions between SNAPIN and M1, M2, and NS2. As a proof of concept for the screening method, we focused on validating the SNAPIN-M1 interaction, which we confirmed through Co-IP and GST pull-down assays, providing the first direct evidence of molecular interplay between SNAPIN and M1. To define the precise interaction interface, future studies will generate truncated mutants of both SNAPIN and virus-encoded proteins to map essential binding domains. Understanding this mechanism will be critical for clarifying the role of SNAPIN in the IAV life cycle. Collectively, these results demonstrate the robustness of our screening strategy and highlight its value for rapidly identifying host–virus protein interactions. Our findings lay the groundwork for future mechanistic studies into how SNAPIN regulates IAV replication.

## 5. Conclusions

In this study, we established a robust live-cell screening platform based on the NanoBiT luciferase complementation system, enabling the rapid and sensitive identification of novel host–virus PPIs. Utilizing this approach, we successfully identified several previously unreported interactions between the host protein SNAPIN and multiple IAV proteins, including M1, M2, and NS2. Among these, the interaction between SNAPIN and the viral matrix protein M1 was further rigorously validated through independent biochemical assays, including co-immunoprecipitation and GST pull-down experiments, confirming both the specificity and reliability of our screening platform. These findings not only underscore the effectiveness of the NanoBiT system in uncovering previously uncharacterized virus–host interactions but also provide new mechanistic insights into the lifecycle of IAV. Our work sets a solid foundation for further functional studies aimed at elucidating the precise role of SNAPIN in the IAV replication cycle. Future research will focus on delineating the biological significance of these interactions and exploring their implications for viral pathogenesis and host adaptation.

## Figures and Tables

**Figure 1 biology-14-01770-f001:**
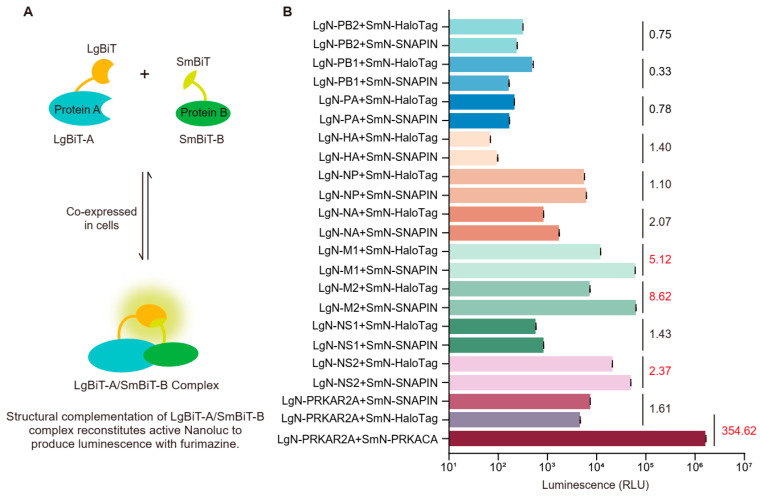
Screening for viral proteins interacting with SNAPIN. (**A**) Schematic of the NanoBiT PPIs assay. Target proteins are fused to the LgBiT or SmBiT fragments. The fusion proteins are co-expressed in cells. Binding between the target proteins enables structural complementation of the LgBiT and SmBiT fragments. The complementation of LgBiT-A/SmBiT-B complex reconstitutes functional NanoLuc luciferase, generating a luminescent signal upon furimazine substrate addition. (**B**) Screening for interactions between SNAPIN and IAV-encoded proteins using the NanoBiT system. 293T cells were co-transfected with plasmid pairs encoding SNAPIN-SmBiT and individual viral protein-LgBiT fusions. Luminescence was measured 24 hpt. Results are shown in relative luminescence units (RLU) with the corresponding fold-change over the negative control; values for M1, M2, and NS2 are highlighted in red. An elevated fold-increase is indicative of a potential interaction, as per the manufacturer’s protocol. Data is representative of three independent experiments performed in triplicate.

**Figure 2 biology-14-01770-f002:**
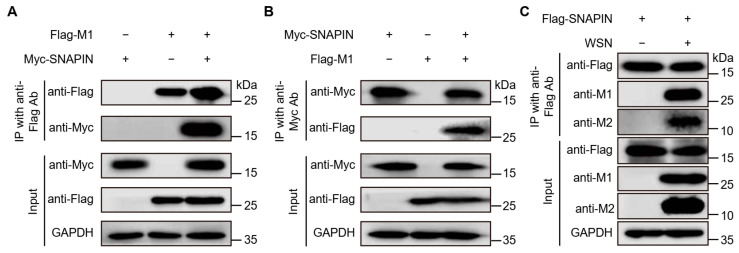
Interaction between SNAPIN and M1 protein using Co-IP. In panel (**A**,**B**), HEK293T cells were transfected with plasmids encoding the indicated proteins. After 24 hpt, cell lysates were subjected to immunoprecipitation using anti-Flag VHH magnetic beads (**A**) or anti-Myc VHH magnetic beads (**B**). The input cell lysates and immunoprecipitated samples were analyzed using Western blotting with the indicated antibodies. The A549 cells stably expressing Flag-tagged SNAPIN (A549-Flag-SNAPIN) were infected with WSN for 24 h. Cell lysates were immunoprecipitated using anti-Flag VHH magnetic beads. Cell lysates and immunoprecipitated samples were analyzed using Western blotting with anti-M1 mAb antibody and anti-M2 mAb (**C**), respectively. Data is representative of three independent experiments performed in triplicate. Raw data of this figure can be found in Appendix A.

**Figure 3 biology-14-01770-f003:**
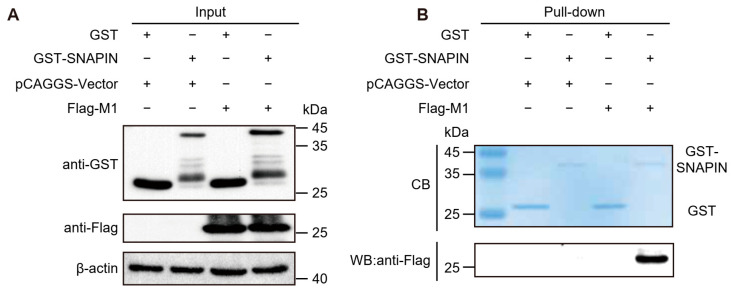
GST pull-down assay of SNAPIN and M1. (**A**) Western blotting analysis of *E. coli* BL21(DE3) expressed purified GST or GST-SNAPIN alongside purified Flag-M1 protein as input samples. (**B**) GST pull-down assays were performed using equal amounts of purified GST or GST-SNAPIN bound to Mag-Beads. Coomassie Blue staining of bead-bound fractions confirms equivalent loading of GST or GST-SNAPIN in the reactions. Western blot analysis of pulled-down complexes using anti-Flag antibody. Data is representative of three independent experiments performed in triplicate. Raw data of this figure can be found in Appendix A.

## Data Availability

The data supporting the conclusions of this article will be available from the corresponding author upon request.

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
