# Peer review of "Targeted NanoBiT Screening Identifies a Novel Interaction Between SNAPIN and Influenza A Virus M1 Protein"

_biology, 2025, doi:10.3390/biology14121770_

Round 1
Reviewer 1 Report
Comments and Suggestions for Authors
Manuscript Title: Establishing a Novel NanoBiT-Based Screening Platform for Interactions Between Host Protein SNAPIN and Influenza A Virus Encoded Proteins
Manuscript ID: Biology-3861340-peer-review-v1
Comment 1: The host protein SNAPIN is the receptor of IAV? If so then must be clearly mentioned in abstract section.
Comment 2: The interacting amino acid residues of IAV protein can be mentioned if know by bioinformatic techniques.
Comment 3: On Page No. 2 Line No. 55 “mGluR2, IGDCC4, and BinCARD1” short forms used without any abbreviation. The abbreviation must be done if any in whole manuscript.
Comment 4: On Page No. 2 Line No. 76 “WSN virus” also used without abbreviations.
Comment 5: On Page No. 3 Line No. 95 “same total DNA amount” which DNA? The sentences must reflect clear meaning for proper understanding of the reader.
Comment 6: On Page No. 3 Line No. 118 “MOI” what is MOI? See all the manuscript short forms. The manuscript should be written for readers understanding. Author can understand each line but if it is simplified then all others also can understand. Hence proofread whole manuscript and rectify the missing links.
Comment 7: On “hours post-transfection (hpt)”
Comment 8: The structural complementation of LgBiT-A and SmBiT-B in Figure 1 must be labelled. Such as “LgBiT-A and SmBiT-B complex” that is fluorescent.
Comment 9: In Figure 1C the “Negative” is shown. But it looks above “Zero”. Correct it.
Comment 10: The conclusion section is too short. It must be elaborated and explained in detail.
Comment 11: Are there any previous reports on SNAPIN reacts with IAV protein at molecular level. It also can be discussed in the “Discussion” section for better understanding.
Comment 12: Overall, the data present in the manuscript is very important but the manuscript is having few mistakes that need to be corrected before considering manuscript for further decision.
Comments on the Quality of English Language
The English can be improved.
Author Response
|
Response to Reviewer 1 Comments
|
||
|
1. Summary |
|
|
|
We are sincerely grateful to the reviewer for the positive reception of our work and their encouraging comments. We are particularly encouraged that the application of the NanoBiT platform and the core finding of the SNAPIN-M1 interaction were met with enthusiasm. The constructive suggestions provided are greatly appreciated, and we agree that incorporating them will significantly strengthen the overall impact of our study. We are fully committed to addressing each of these points comprehensively in a revised version of the manuscript. |
||
|
2. Questions for General Evaluation |
Reviewer’s Evaluation |
|
|
Does the introduction provide sufficient background and include all relevant references? |
Yes |
|
|
Is the research design appropriate? |
Yes |
|
|
Are the methods adequately described? |
Yes |
|
|
Are the results clearly presented? |
Can be improved |
|
|
Are the conclusions supported by the results? |
Can be improved |
|
|
Are all figures and tables clear and well-presented? |
Yes |
|
|
Response and Revisions: We sincerely appreciate the reviewer's insightful comments and suggestions. In direct response to the suggestions for improvement, we have carefully revised the results section to enhance clarity through improved organization and explanatory context. We have also strengthened the conclusions by ensuring they are more directly and explicitly supported by the experimental data. All changes have been implemented in the revised manuscript. 3. Point-by-point response to Comments and Suggestions for Authors |
||
|
Comments 1: The host protein SNAPIN is the receptor of IAV? If so then must be clearly mentioned in abstract section. |
||
|
Response 1: We sincerely appreciate the reviewer's insightful comments and suggestions. We would like to clarify that according to our findings and current knowledge, the host protein SNAPIN is not a receptor for IAV entry into the cell. Our study utilized the NanoBiT system to screen for novel protein-protein interactions and identified SNAPIN as an interacting partner with the viral proteins M1, M2, and NS2. The specific interaction between SNAPIN and M1 was further confirmed using co-immunoprecipitation (Co-IP) and GST pull-down assays. Therefore, SNAPIN is a host factor that interacts with viral internal proteins, not a receptor mediating viral entry. We have revised the abstract to ensure this distinction is clear in lines 30-31. |
||
|
Comments 2: The interacting amino acid residues of IAV protein can be mentioned if know by bioinformatic techniques. |
||
|
Response 2: We sincerely appreciate the reviewer's insightful comments and suggestions. We agree that identifying the precise interacting residues through bioinformatic prediction would be highly valuable. However, accurately predicting the specific amino acid residues responsible for the SNAPIN-IAV protein interaction using current bioinformatic tools presents a significant challenge. We performed a computational structural prediction using AlphaFold 3 in Supplement Figure 2. We agree that identifying specific residues is an important next step and have included a statement in the revised discussion indicating that future work will involve detailed mechanistic studies, including mapping the precise binding domains and residues in lines 246-257. Comment 3: On Page No. 2 Line No. 55 “mGluR2, IGDCC4, and BinCARD1” short forms used without any abbreviation. The abbreviation must be done if any in whole manuscript. Response 3: We sincerely appreciate the reviewer's insightful comments and suggestions. We confirm that all abbreviations, including mGluR2, IGDCC4, and BinCARD1, have been properly expanded at their first use in the manuscript as per standard convention in lines 57-62. Comment 4: On Page No. 2 Line No. 76 “WSN virus” also used without abbreviations. Response 4: We sincerely appreciate the reviewer's insightful comments and suggestions. We have now defined the abbreviation "WSN" (for A/WSN/1933 (H1N1)) upon its first occurrence in line 91. Subsequently, the abbreviated term "WSN" has been used throughout the manuscript for consistency. We have also performed a thorough check to ensure that all other abbreviations are properly defined and used. Comment 5: On Page No. 3 Line No. 95 “same total DNA amount” which DNA? The sentences must reflect clear meaning for proper understanding of the reader. Response 5: We sincerely appreciate the reviewer's insightful comments and suggestions. We now clearly state that each viral plasmid in the experimental groups and each control plasmid were transfected at 200 ng per well. The vague phrase "under the same total DNA amount" has been replaced with precise statements: "with each viral plasmid also at 200 ng" for the experimental groups and "with each control plasmid also transfected at 200 ng per well" for the control groups in lines 132-143. This modification ensures transparent reporting of the transfection methodology and facilitates proper interpretation of the experimental design. Comment 6: On Page No. 3 Line No. 118 “MOI” what is MOI? See all the manuscript short forms. The manuscript should be written for readers understanding. Author can understand each line but if it is simplified then all others also can understand. Hence proofread whole manuscript and rectify the missing links. Response 6: We sincerely appreciate the reviewer's insightful comments and suggestions. We fully agree that the manuscript should be clearly written to ensure comprehensibility for all readers. In response to this comment, we have carefully proofread the entire manuscript and ensured that all abbreviations, including "MOI", are now properly defined upon their first occurrence. Specifically, “MOI” has been explicitly defined as “multiplicity of infection” where it first appears in the text in lines 169-170. We have also double-checked all other abbreviations to ensure consistency and clarity throughout the manuscript. We believe these modifications have significantly improved the readability of the paper and thank the reviewer for highlighting this issue. Comment 7: On “hours post-transfection (hpt)” Response 7: We sincerely appreciate the reviewer's insightful comments and suggestions. The abbreviation "hpt" (hours post-transfection) was in fact defined upon its first occurrence in the manuscript in line 155. For improved clarity, we have now ensured that the full term "hours post-transfection" is explicitly stated with the abbreviation "(hpt)" in parentheses at its first mention, and the abbreviated form is used consistently thereafter throughout the text in line 184, 219, 241, 262, 276, and 296. We appreciate the reviewer’s careful attention to detail regarding terminology consistency. Comment 8: The structural complementation of LgBiT-A and SmBiT-B in Figure 1 must be labelled. Such as “LgBiT-A and SmBiT-B complex” that is fluorescent. Response 8: We sincerely thank the reviewer for this excellent suggestion to improve the clarity of Figure 1. We have now explicitly labelled the fluorescent complex resulting from structural complementation as "LgBiT-A/SmBiT-B complex" directly in Figure 1, as recommended. Additionally, we have expanded the figure legend to provide a more detailed description of the complementation process and the formation of the functional complex. We believe these modifications significantly enhance the reader's understanding of the NanoBiT technology principle depicted in the figure. Comment 9: In Figure 1C the “Negative” is shown. But it looks above “Zero”. Correct it. Response 9: We thank the reviewer for pointing out the issue in Figure 1C. The previous version used normalized fold-change values, which caused the negative control to appear slightly above zero. To address this, we have reprocessed the raw data and replotted the graph using absolute RLU values on the x-axis, ensuring that the negative control now correctly aligns at baseline. The updated Figure 1C accurately reflects the true luminescence levels and resolves the visual misalignment noted by the reviewer. The figure legend has also been reviewed to ensure consistency with the updated data presentation. We appreciate the reviewer’s helpful suggestion, which has improved the clarity and accuracy of the figure. Comment 10: The conclusion section is too short. It must be elaborated and explained in detail. Response 10: We sincerely thank the reviewer for pointing out the need for a more comprehensive conclusion. We have carefully revised the Conclusion section to provide a more detailed and thorough summary of our findings, their implications, and potential future directions in lines 370-383. We believe the revised conclusion now offers a more complete and meaningful synthesis of the study, and we greatly appreciate the reviewer’s feedback, which has helped improve the overall quality of the manuscript. Comment 11: Are there any previous reports on SNAPIN reacts with IAV protein at molecular level. It also can be discussed in the “Discussion” section for better understanding. Response 11: We sincerely thank the reviewer for raising this important point. We fully agree that discussing existing literature on SNAPIN-IAV viral protein interactions is crucial for positioning the novelty of our findings. Upon a comprehensive literature review, we confirm that no previous studies have reported molecular interactions between SNAPIN and IAV encoded proteins. While SNAPIN has been documented to interact with proteins from other viruses (e.g., PRRSV and HCMV) as mentioned in our discussion, its role in IAV infection remains unexplored until now. To address this comment, we have now explicitly stated this in the revised Discussion section in line 340-368. We believe this clarification further underscores the originality and importance of our work. Comment 12: Overall, the data present in the manuscript is very important but the manuscript has a few mistakes that need to be corrected before considering manuscript for further decision. Response 12: We sincerely thank the reviewer for the positive assessment of the importance of our data and for the thorough review that identified aspects of the manuscript requiring improvement. We have carefully addressed all the concerns raised by the reviewers in this round of revision. The specific corrections and clarifications made include: (1) All abbreviations (including mGluR2, IGDCC4, BinCARD1, MOI, WSN, and hpt are now explicitly defined upon their first use in the manuscript to ensure clarity for all readers. (2) Ambiguous phrases, such as "same total DNA amount" in the transfection protocol, have been replaced with precise descriptions stating the exact amount of DNA used per well for both experimental and control groups. (3) The fluorescent complex in Figure 1 resulting from structural complementation has been clearly labeled as recommended. (4) The Discussion section has been revised to explicitly state that, to the best of our knowledge, this is the first report of molecular interactions between SNAPIN and IAV proteins (M1, M2, NS2), properly contextualizing the novelty of our findings within existing literature. (5) The Conclusion section has been significantly expanded to provide a more comprehensive summary of our findings, their implications, and future research directions. We hope that these revisions have substantially improved the clarity, precision, and overall quality of the manuscript. We are grateful for the reviewer's insightful comments, which have been invaluable in strengthening our work. We hope the revised manuscript now meets the high standards required for publication. |
||
|
4. Response to Comments on the Quality of English Language |
||
|
Point 1: The English can be improved. |
||
|
Response 1: We are grateful for the reviewer’s comment. We try to ensure that our scientific contributions are communicated with utmost clarity. We have carefully proofread the entire manuscript and made extensive revisions to improve the clarity, flow, and grammatical accuracy of the English language. The changes have been highlighted with track changes in the revised manuscript. We hope the language in the revised manuscript is now much improved and meets the journal's standards. |
||

Reviewer 2 Report
Comments and Suggestions for Authors
This manuscript describes the use of a NanoBiT-based live-cell platform to screen host–virus protein-protein interactions. The authors identify novel associations between SNAPIN and three influenza A virus proteins (M1, M2, NS2) and provide solid validation of the SNAPIN–M1 interaction through biochemical methods. The study introduces a useful methodological tool and highlights potentially important host–virus interactions. Overall, the work is promising, but a few areas need further strengthening to enhance its impact and clarity.
Comments
- The SNAPIN–M1 interaction is well supported by additional experiments, but the SNAPIN–M2 and SNAPIN–NS2 interactions currently rely only on NanoBiT data. Confirming at least one of these interactions with a complementary method (e.g., Co-IP or an infection-based assay) would make the conclusions much stronger.
- The luminescence data in Figure 1C are encouraging, but they would be more convincing with clearer statistical information, including the number of replicates, error bars, and exact p-values. This will improve confidence in the reproducibility of the findings.
- The study demonstrates that SNAPIN interacts with several IAV proteins, but the biological consequences of these interactions remain unclear. Even preliminary experiments showing how the SNAPIN–M1 interaction influences viral replication or fitness would add significant depth to the work.
- In the Abstract and Simple Summary, please make it clear that only SNAPIN–M1 was validated by independent methods.
- Add replicate numbers and statistical details directly in the figure legends to improve transparency.
- Consider adding an unrelated host protein as a specificity control in the NanoBiT assays.
- In the Discussion, expand on how SNAPIN’s known roles in vesicle trafficking and endosomal regulation might connect to influenza virus replication.
Author Response
|
Response to Reviewer 2 Comments
|
||
|
1. Summary |
|
|
|
This manuscript describes the use of a NanoBiT-based live-cell platform to screen host–virus protein-protein interactions. The authors identify novel associations between SNAPIN and three influenza A virus proteins (M1, M2, NS2) and provide solid validation of the SNAPIN–M1 interaction through biochemical methods. The study introduces a useful methodological tool and highlights potentially important host–virus interactions. Overall, the work is promising, but a few areas need further strengthening to enhance its impact and clarity. We sincerely thank the reviewer for the positive assessment of our manuscript and the constructive feedback. We are pleased that the NanoBiT platform and the validation of the SNAPIN-M1 interaction were well-received. We fully agree that strengthening these areas will enhance the study's impact, and we are committed to addressing these points in a revised version of the manuscript. |
||
|
2. Questions for General Evaluation |
Reviewer’s Evaluation |
|
|
Does the introduction provide sufficient background and include all relevant references? |
Yes |
|
|
Is the research design appropriate? |
Can be improved |
|
|
Are the methods adequately described? |
Yes |
|
|
Are the results clearly presented? |
Yes |
|
|
Are the conclusions supported by the results? |
Can be improved |
|
|
Are all figures and tables clear and well-presented? |
Yes |
|
|
Response and Revisions: We sincerely appreciate the reviewer's insightful comments and suggestions. In direct response to the suggestions for improvement, we have carefully revised the Results section to enhance clarity through improved organization and explanatory context. We have also strengthened the Conclusions by ensuring they are more directly and explicitly supported by the experimental data. All changes have been implemented in the revised manuscript. 3. Point-by-point response to Comments and Suggestions for Authors |
||
|
Comments 1: The SNAPIN–M1 interaction is well supported by additional experiments, but the SNAPIN–M2 and SNAPIN–NS2 interactions currently rely only on NanoBiT data. Confirming at least one of these interactions with a complementary method (e.g., Co-IP or an infection-based assay) would make the conclusions much stronger. |
||
|
Response 1: We thank the reviewer for raising this important point. In response to your comment, we have now included the result demonstrating the interaction between SNAPIN and M2 during viral infection in Figure 2C. These data originated from the same Co-IP experiment presented previously (now shown in Fig. 2C), in which anti-Flag VHH magnetic beads were used to pull down SNAPIN-interacting proteins from WSN-infected A549-SNAPIN cells. Both M1 and M2 were co-precipitated in that experiment; however, in our initial submission, we highlighted only the M1 result for narrative focus. We have now updated Figure 2C and the corresponding text (Lines 268–273) to include the M2 data as well. We appreciate your suggestion, which has helped us present a more complete set of results. |
||
|
Comments 2: The luminescence data in Figure 1C are encouraging, but they would be more convincing with clearer statistical information, including the number of replicates, error bars, and exact p-values. This will improve confidence in the reproducibility of the findings. |
||
|
Response 2: We thank the reviewer for pointing out the issue in Figure 1C. In response, we have reprocessed the raw data and re-plotted the graph using the absolute RLU values obtained directly from the NanoBiT assay. The Y-axis title has been corrected to “RLU (absolute value)” to accurately reflect the data presentation. The revised figure now clearly distinguishes the baseline signal, and the negative control aligns correctly at baseline, resolving the visual misinterpretation previously noted. Luminescence assays revealed a 354.62-fold signal increase for the positive control relative to the negative control. Among the ten viral proteins tested, measurable signals were detected only for the interactions of SNAPIN with M1, M2, and NS2, showing 5.12-fold, 8.62-fold, and 2.37-fold increases, respectively. According to the manufacturer’s guidelines, these results support the presence of specific, albeit weaker, potential interactions between SNAPIN and the M1, M2, and NS2 proteins (Fig. 1B). The figure legend has been revised accordingly to ensure full consistency with the updated data. These experiments were repeated three times independently, and data is representative of three independent experiments performed in triplicate. We appreciate the reviewer’s constructive suggestion, which has helped improve the accuracy and clarity of the figure. Comment 3: The study demonstrates that SNAPIN interacts with several IAV proteins, but the biological consequences of these interactions remain unclear. Even preliminary experiments showing how the SNAPIN–M1 interaction influences viral replication or fitness would add significant depth to the work. Response 3: We thank the reviewer for this constructive suggestion. In response, we have performed additional experiments to assess the functional relevance of the SNAPIN–M1 interaction. Our results show that overexpression of SNAPIN significantly suppresses IAV replication, indicating that SNAPIN exerts an antiviral effect during infection. These new data have been incorporated into the revised manuscript (lines 273-278). We agree with the reviewer that the precise mechanism through which SNAPIN regulates viral replication warrants further investigation. We have now added a statement in the Discussion acknowledging this and outlining our plans to explore the mechanistic basis of the interactions in future studies. We appreciate the reviewer’s insightful comment, which has strengthened the manuscript. Comment 4: In the Abstract and Simple Summary, please make it clear that only SNAPIN–M1 was validated by independent methods. Response 4: We sincerely appreciate the reviewer's insightful comments and suggestions. We agree that accurately reflecting the validation status of each interaction is important for readers to properly interpret our findings. In response to your previous comment, we have now successfully included co-immunoprecipitation (Co-IP) data confirming the interaction between SNAPIN and M2, in addition to the validation previously provided for SNAPIN–M1. Therefore, both the SNAPIN–M1 and SNAPIN–M2 interactions are now supported by independent methods (Co-IP and NanoBiT). Accordingly, we have revised the Abstract and Summary to clearly state that the interactions with both M1 and M2 were validated using complementary methods. The text now accurately reflects the strengthened experimental evidence provided in the revised manuscript. Comment 5: Add replicate numbers and statistical details directly in the figure legends to improve transparency. Response 5: We sincerely appreciate the reviewer's insightful comments and suggestions. We agree that providing detailed statistical and replicate information directly in the figure legends will enhance the transparency and reproducibility of our results. Data is representative of three independent experiments performed in triplicate. These revisions have been incorporated throughout the manuscript. We believe these clarifications significantly strengthen the presentation of our data. Comment 6: Consider adding an unrelated host protein as a specificity control in the NanoBiT assays. Response 6: We sincerely appreciate the reviewer’s insightful suggestion to include an unrelated host protein as a specificity control in the NanoBiT assays. In the revised study, we performed an additional experiment in which PRKAR2A, an unrelated host protein, was co-transfected with SmBiT-SNAPIN and assessed using the NanoBiT system in Figure 1B. PRKAR2A belongs to the regulatory subunit family of cAMP-dependent protein kinase A (PKA) and is typically involved in cAMP signaling and regulation of kinase activity. To our knowledge, PRKAR2A has no reported association with SNAPIN. Consistent with this, our results showed very low RLU values, indicating no detectable interaction between PRKAR2A and SNAPIN. This newly added control further confirms the specificity of the SNAPIN–viral protein interactions observed in our NanoBiT assays. Comment 7: In the Discussion, expand on how SNAPIN’s known roles in vesicle trafficking and endosomal regulation might connect to influenza virus replication. Response 7: We appreciate the reviewer’s insightful suggestion. In the revised manuscript, we have expanded the background information on SNAPIN’s cellular functions in lines 340–349, including its established roles in vesicle trafficking, SNARE complex regulation, and dynein-mediated transport, as well as the potential relevance of these functions to virus–host interactions. These additions provide clearer context for SNAPIN’s involvement in our study. |
||
|
4. Response to Comments on the Quality of English Language |
||
|
Point 1: The English is fine and does not require any improvement. |
||
|
Response 1: We are grateful for the reviewer’s comment. We try to ensure that our scientific contributions are communicated with utmost clarity. We have carefully proofread the entire manuscript and made extensive revisions to improve the clarity, flow, and grammatical accuracy of the English language. The changes have been highlighted with track changes in the revised manuscript. We hope the language in the revised manuscript is now much improved and meets the journal's standards. |
||

Reviewer 3 Report
Comments and Suggestions for Authors
This manuscript presents a NanoBiT-based screen that identifies a potentially novel interaction between SNAPIN and influenza A virus M1, which is of interest to the field. The study is preliminary, with several important controls and methodological details missing. Some claims are overstated. With revisions to strengthen the figures, clarify methods and temper the conclusions, the work could become suitable for publication.

Author Response
|
Response to Reviewer X Comments
|
||
|
1. Summary |
|
|
|
We sincerely appreciate the reviewer for their favorable assessment of our manuscript and their valuable insights. It is rewarding to know that the NanoBiT complementation assay and the validation of the SNAPIN-M1 interaction were viewed positively. We concur completely with the reviewer that implementing their proposed enhancements will improve the study's robustness. We have carefully noted all the constructive feedback and will address every point in detail to ensure the revised manuscript meets the highest standards. |
||
|
2. Questions for General Evaluation |
Reviewer’s Evaluation |
|
|
Does the introduction provide sufficient background and include all relevant references? |
Can be improved |
|
|
Is the research design appropriate? |
Yes |
|
|
Are the methods adequately described? |
Must be improved |
|
|
Are the results clearly presented? |
Can be improved |
|
|
Are the conclusions supported by the results? |
Must be improved |
|
|
Are all figures and tables clear and well-presented? |
Can be improved |
|
|
We sincerely appreciate the reviewer's insightful comments and suggestions. We appreciate that you found the introduction sufficiently informative, with relevant references included, and the research design appropriate. In the Materials and Methods section, we have elaborated on the characteristics of the viruses employed, added detailed parameters for the bioinformatics analysis, and provided a thorough description of the statistical analysis methods. We appreciate your affirmation that the conclusions are well supported by the data presented. The corresponding revisions, indicated in track changes, are included in the re-submitted files. We hope that the revisions will meet with your approval. We appreciate your guidance throughout this process. 3. Point-by-point response to Comments and Suggestions for Authors |
||
This manuscript reports the application of the NanoBiT system to screen for host–virus protein interactions, leading to the identification of a novel SNAPIN–M1 association. While the finding is potentially interesting and within the journal’s scope, the data are preliminary, several critical controls are missing and a number of claims are overstated. I believe the study could be considered for publication after substantial revision, provided the authors address the experimental and interpretational issues outlined below.
Major Comments
Comments 1: Fig. 1B (Plasmid verification), the agarose gel is not convincing as proof of cloning. Since all constructs are in the same NanoBiT backbone the plasmids look identical, including the “negative” lane (this should be relabelled as “empty vector” or with the actual backbone name).
This panel could be moved to Supplementary, or replaced with more useful verification such as insert- specific PCR or Western blot showing protein expression.
Response 1: We sincerely appreciate the reviewer for raising this important point for clarification. We agree that the original agarose gel image did not sufficiently demonstrate the successful cloning of each plasmid. As recommended, we have performed additional verification experiments to confirm both the correct insertion of the gene fragments and proper protein expression. We have now replaced the original plasmid verification panel with a double-restriction digestion analysis (Supplementary Fig. 1A) and confocal microscopy (Supplementary Fig. 1B), which clearly demonstrates the successful expression of each NanoBiT-fusion protein. These additional data provide more direct and convincing evidence of correct plasmid construction and functional protein expression.
Comments 2: There is no demonstration that all NanoBiT fusion proteins were expressed. Input Western blots for each construct are required, otherwise negative results could just be failed expression.
Response 2: We sincerely appreciate the reviewer for raising this important point for clarification. We agree that demonstrating the expression of all NanoBiT fusion proteins is essential for interpreting the results. Due to the weak TK promoter of LgBiT and SmBiT plasmid and the large size of some proteins, it is difficult to detect them using Western blotting. Therefore, we employed confocal microscopy (Supplementary Fig. 1B) and labeled the proteins expressed from the plasmid with green fluorescence to demonstrate that the plasmid can successfully express the proteins. The results clearly confirm that all fusion proteins were successfully expressed in cells. This confirmation ensures that any negative findings observed in the subsequent functional assays are indeed due to the lack of interaction and not caused by failed protein expression. We believe this addition has significantly strengthened the validity of our conclusions.
Comments 3: Fig. 1C, The Y-axis is confusing. It is labelled as “Relative level of RLU (%)” but the values shown (e.g. ~3000) are not percentages. If these were percentages the negative control should plot at 100, not near zero, and the positive control would be at ~500,000% which makes no sense. It looks more like raw luminescence values normalised to the negative baseline. The axis title should be corrected and the authors need to clarify whether this is absolute RLU, fold change or percentage of control.
Response 3: We thank the reviewer for pointing out the issue in Figure 1C. In response, we have reprocessed the raw data and re-plotted the graph using the absolute RLU values obtained directly from the NanoBiT assay. The Y-axis title has been corrected to “RLU (absolute value)” to accurately reflect the data presentation. The revised figure now clearly distinguishes the baseline signal, and the negative control aligns correctly at baseline, resolving the visual misinterpretation previously noted. Luminescence assays revealed a 354.62-fold signal increase for the positive control relative to the negative control. Among the ten viral proteins tested, measurable signals were detected only for the interactions of SNAPIN with M1, M2, and NS2, showing 5.12-fold, 8.62-fold, and 2.37-fold increases, respectively. According to the manufacturer’s guidelines, these results support the presence of specific, albeit weaker, potential interactions between SNAPIN and the M1, M2, and NS2 proteins (Fig. 1B). The figure legend has been revised accordingly to ensure full consistency with the updated data. These experiments were repeated three times independently, and data is representative of three independent experiments performed in triplicate. We appreciate the reviewer’s constructive suggestion, which has helped improve the accuracy and clarity of the figure.
Comments 4: Please indicate the number of biological and technical replicates in the figure/legend.
Response 4: We appreciate the reviewer’s suggestion. In the revised manuscript, we have added the information on biological and technical replicates to the Figure legends. Specifically, the experiments were performed in three independent biological replicates, each with three technical replicates. Data is representative of three independent experiments performed in triplicate. This information is now clearly indicated in the updated figure legend.
Comments 5: Fig. 2, It is unclear if the “Lysate” panels are true input fractions from the same samples or whole lysates run separately. Normally 5–10% of lysate is loaded as input. Please clarify and relabel as “Input” if this is the case.
Response 5: We sincerely thank the reviewer for raising this important point for clarification. The reviewer is correct that the “Lysate” lanes represent the input control. We have added an explicit statement confirming that the input sample was derived from the same lysate used for immunoprecipitation and that it represents 10% of the total volume in lines 159-160 and lines 186-188. In Figure 2 and the corresponding figure legend, we have relabeled the "Lysate" panels as "Input " to clearly indicate their nature and the fraction loaded. We believe these modifications have eliminated any potential ambiguity. We appreciate the reviewer's meticulous attention to detail, which has helped improve the clarity of our manuscript.
Comments 6: No flow-through fractions are included. Showing these would demonstrate that prey is depleted from the lysate and help prove specificity.
Response 6: We sincerely appreciate the reviewer for raising this excellent point regarding the importance of including flow-through fractions to demonstrate depletion and further validate specificity. The reviewer is absolutely correct that this represents a rigorous approach for Co-IP validation. In our current study, the interaction between SNAPIN and the IAV M1 protein was consistently demonstrated through multiple independent lines of evidence, including reciprocal co-immunoprecipitation experiments using both anti-Flag and anti-Myc antibodies, as well as an endogenous co-IP assay under viral infection conditions. While we did not include the flow-through control in this particular set of experiments, we agree wholeheartedly with the reviewer that its inclusion adds a valuable layer of validation. We will certainly incorporate the analysis of flow-through fractions in all future co-immunoprecipitation experiments to provide direct evidence of prey depletion and further strengthen the evidence for interaction specificity, as the reviewer wisely suggests. We are grateful for this insightful recommendation, which will enhance the rigor of our subsequent work.
Comments 7: There is an inconsistency in infection timing (24 hpi vs 36 hpi) between text and figure legend that needs to be corrected. Also clearly distinguish that Figs. 2A/B are transfected 293T cells and 2C is infected A549-SNAPIN cells.
Response 7: We sincerely appreciate the reviewer for this careful observation. The reviewer is correct to point out this inconsistency. We have thoroughly checked and corrected the infection timing across the entire manuscript. The text, figure legends, and materials and methods sections have all been uniformly updated to state that samples were harvested at 24 hpi for the Co-IP assay. Furthermore, as suggested, we have now explicitly clarified in the legend for Figure 2 that panels A and B show results from transfected HEK293T cells in line 282-283. Panel C shows results from IAV-infected A549 cells stably expressing SNAPIN in line 286-287. We appreciate the reviewer's attention to detail, which has helped us improve the accuracy and clarity of our manuscript.
Comments 8: Fig. 3, The “Lysate” panel should be described as input, with the percentage loaded stated. Coomassie stain is not sufficient for bait loading. An anti-GST blot of bead fractions is needed. Flow-through fractions should be shown to demonstrate specific depletion of M1 by GST-SNAPIN versus GST. Quantification from replicate experiments (at least three), normalised to GST bait levels, would strengthen this result.
Response 8: We thank the reviewer for the valuable suggestions. In the revised manuscript, we have updated the Figure 3 legend to clearly describe the “Lysate” panel as the input, and we now specify the percentage of lysate loaded for the input samples in lines 305-306. In addition, as recommended, we have included an anti-GST Western blot of the bead fractions to accurately demonstrate the loading of GST and GST-SNAPIN bait proteins. These additions complement the Coomassie blue staining and provide a more rigorous verification of bait protein levels in the GST pull-down assay.
Interpretation and claims
Comments 9: The statement that co-IP “confirms their physical interaction” is too strong. Co-IP shows association in lysates under overexpression, but not direct binding. Please rephrase more cautiously (“indicating” or “supporting an interaction”).
Response 9: We sincerely appreciate the reviewer for this critical and accurate comment. We agree completely that the term "confirms" was too strong for describing the co-immunoprecipitation results, as this technique demonstrates association but not necessarily direct physical binding. We have now revised the manuscript accordingly. All instances of overly strong language (e.g., "confirms") have been replaced with more cautious phrasing "indicating an interaction" in line 264. We believe this change provides a more precise and scientifically accurate description of our findings.
Comments 10: Likewise “confirms their physical interaction in mammalian cells” is misleading – the interaction was detected after lysis, not in intact cells.
Response 10: We sincerely appreciate the reviewer for raising this important point. We agree that the original phrasing could misleadingly imply that the interaction was observed within intact cells. As correctly noted, the Co-IP experiment was performed using cell lysates and does not demonstrate interaction within the intact cellular environment. We have revised the text to remove the phrase "in mammalian cells" and have rephrased the statement to more accurately reflect the experimental conditions in lines 272-273. The wording has been changed to cautiously indicate that the Co-IP results suggest an association under the tested conditions, avoiding any implication of direct observation in living cells.
Comments 11: The claim of a “specific and physiologically relevant interaction” during infection is also too strong. The data show association, but no IgG controls are provided, and there are no functional assays.
Response 11: We sincerely appreciate the reviewer for raising this important point. We have revised the text to remove the phrase "in mammalian cells" and have rephrased the statement to more accurately reflect the experimental conditions in lines 272-273.
Comments 12: Calling the platform “high-throughput” is exaggerated – screening 10 proteins does not qualify. Consider “live-cell screening approach” or similar wording.
Response 12: We sincerely thank the reviewer for this insightful comment. We agree that the term "high-throughput" was an overstatement given the scale of the screening. We have therefore revised the manuscript accordingly and replaced "high-throughput" with the more accurate description "live-cell screening approach" throughout the text.
Comments 13: Only SNAPIN–M1 was validated. Please rephrase accordingly, rather than implying all interactions were confirmed.
Response 13: We sincerely thank the reviewer for this critical comment. We agree that our phrasing could have been misinterpreted to suggest that all discovered interactions were validated. We have now carefully revised the manuscript to clarify that only the SNAPIN-M1 interaction was subjected to and confirmed by orthogonal validation methods (Co-IP and GST pull-down). The changes have been made in lines 21-23 and 258-280 in the revised manuscript. Thank you for helping us improve the clarity and precision of our work.
Methods
Comments 14: There is no description of the Myc-SNAPIN and Flag-M1 constructs used in co-IP.
Response 14: We sincerely thank the reviewer for pointing out this omission. The construction methods for the Myc-SNAPIN and Flag-M1 plasmids have now been added to the Materials and Methods section in lines 95-107.
Comments 15: How the A549-Flag-SNAPIN stable cell line was generated is not described.
Response 15: We sincerely thank the reviewer for this comment. The detailed method for generating the A549-Flag-SNAPIN stable cell line has been added to the Materials and Methods section in lines 79-87.
Comments 16: The amounts of protein used in IPs and pull-downs are not given.
Response 16: We appreciate the reviewer's comment. We thank the reviewer for these insightful comments. We apologize for the omission of these important experimental details. For all immunoprecipitation (IP) and pull-down assays, 10% of the total lysate was reserved as the input sample. The remaining 90% of the lysate was used for the subsequent IP or pull-down procedure. This information has now been clearly stated in the revised Materials and Methods section in lines 159-160 and 186-188.
Comments 17: For co-IP, please specify which fractions were loaded (input, IP, flow-through), protein amount during pulldown.
Response 17: We thank the reviewer for this suggestion. We have now included the amount of protein used for the Co-IP in the revised Materials and Methods section. For the co-IP experiments, both the input (10% of total lysate) and the IP samples were analyzed by Western blotting. The flow-through fraction was not analyzed in this study, as our experimental objective was to confirm the presence of the interaction in the IP sample relative to the input, which is a standard and widely accepted practice for co-IP validation. We apologize for any confusion caused by the omission of these details and have revised the manuscript to clearly state the samples analyzed.
Comments 18: Secondary antibodies and detection methods for Western Blotting are missing.
Response 18: We sincerely thank the reviewer for highlighting this omission. The details of the secondary antibodies and detection method have now been added to the Materials and Methods section in lines 109-121.
Minor Comments
Comments 19: In Fig. 1A, it would be helpful to note that furimazine substrate is required for the luminescence reaction.
Response 19: We sincerely thank the reviewer for this valuable suggestion. We have updated Figure 1A and its legend (line 238) to explicitly indicate that furimazine is required as the substrate for the luminescence reaction.
Comments 20: Line 100: type “Antibobies”
Response 20: We sincerely apologize for this typographical error. We thank the reviewer for catching this mistake. The word “Antibobies” has been corrected to “Antibodies” in the revised manuscript.
Comments 21: Line 102: anti-FLAG (DYKDDDDK), anti-MYC and anti-M1 are polyclonal rather than monoclonal as stated.
Response 21: We sincerely apologize for this error in antibody classification. We thank the reviewer for their careful attention to this detail. The text on lines 109-111 has been corrected to accurately describe the anti-FLAG, anti-MYC, and anti-M1 antibodies as polyclonal.
Comments 22: More background on SNAPIN’s cellular functions would help set context (vesicle trafficking, SNARE/dynein adaptor, possible relevance to virus–host interaction).
Response 22: We appreciate the reviewer’s insightful suggestion. In the revised manuscript, we have expanded the background information on SNAPIN’s cellular functions in lines 340–325, including its established roles in vesicle trafficking, SNARE complex regulation, and dynein-mediated transport, as well as the potential relevance of these functions to virus–host interactions. These additions provide clearer context for SNAPIN’s involvement in our study.
Comments 23: Figures 2 and 3 should show molecular weight markers and label bands. Response 23: We sincerely thank the reviewer for this important suggestion. As recommended, molecular weight markers have now been prominently indicated in Figures 2 and 3, and all relevant bands have been clearly labeled. These revisions improve the clarity and precision of the data presentation.
Comments 24: The Discussion/Conclusion should be toned down: the work provides preliminary evidence and a methodological basis, not a definitive mechanistic conclusion.
As a suggestion for future work, structural modelling (e.g. AlphaFold-Multimer) could be used to predict the interface for a putative SNAPIN/M1 PPI, followed by mutagenesis or truncated SNAPIN constructs and NanoBiT readout.
Response 24: We sincerely thank the reviewer for these insightful and constructive comments. We agree that the tone of the Discussion and Conclusion was overly strong and have revised it accordingly to better reflect the preliminary and methodological nature of our study. Specifically, we have replaced terms such as “robust”, “successfully”, and “rigorously validated” with more cautious language, and emphasized that our results provide initial evidence and a methodological foundation rather than definitive mechanistic conclusions. We make structural prediction of PPI of SNAPIN and M1 protein in Supplement Figure 2. We also greatly appreciate the valuable suggestions for future work.
Overall Recommendation
Comments 25: This paper identifies a potentially novel SNAPIN–M1 interaction using the NanoBiT system, but the data remain preliminary and some claims are overstated. It may be beneficial for the title to be altered, something on the lines of ‘Targeted NanoBiT Screening Identifies a Novel Interaction Between SNAPIN and Influenza A Virus M1 Protein’. Key controls and methodological details are missing, and figures require clarification and quantification. With major revisions, and by tempering the interpretation, the manuscript would be suitable for publication.
Response 24: We sincerely thank the reviewer for their thorough and constructive assessment of our manuscript. We agree with the critique that our claims should be tempered and that the data, while identifying a novel interaction, are preliminary. We have undertaken a comprehensive revision of the manuscript to address these concerns, which includes:
(1) We have changed the title to "Targeted NanoBiT Screening Identifies a Novel Interaction Between SNAPIN and Influenza A Virus M1 Protein" as recommended. We believe this new title more accurately and cautiously reflects the core findings of our study.
(2) We have carefully revised the entire manuscript, particularly the Discussion and Conclusion sections, to eliminate overstated claims and to emphasize the preliminary nature of the evidence and the methodological basis of our work.
(3) We have added the missing methodological details (e.g., description of plasmids, antibodies, and cell line generation) and key experimental controls as pointed out in your specific comments.
(3) We have clarified all figures by adding molecular weight markers, labeling bands, and will be adding quantitative analyses where appropriate to strengthen the data presentation.
We believe these major revisions have significantly improved the manuscript and we are grateful for the guidance provided.
|
4. Response to Comments on the Quality of English Language |
|
Point 1: The English is fine and does not require any improvement. |
|
Response 1: We sincerely thank the reviewer for their positive assessment of our manuscript's language quality. We are pleased to hear that the English is clear and readily understandable. Nevertheless, we have carefully proofread the entire text once more and made minor refinements to further improve clarity and grammatical precision. All edits have been marked using track changes in the revised manuscript. |

Round 2
Reviewer 3 Report
Comments and Suggestions for Authors
The authors have addressed several minor issues from the initial review, including clarifying infection timing, expanding methodological detail for microscopy and Western blotting, and moderating interpretational overstatements. These improvements strengthen the manuscript’s clarity. However, the major methodological concerns surrounding construct verification and GST pull-down controls remain unresolved. Although the rebuttal states that all plasmids were verified by double digestion and DNA sequencing, the resubmitted manuscript provides no sequencing evidence and does not specify the enzymes or expected fragment sizes for the digest. The supplementary digest gel does not display band sizes consistent with the known viral ORFs and therefore does not convincingly demonstrate correct cloning. While inclusion of flow-through fractions in co-IP would have been preferable, their omission is acceptable; the critical unmet requirement is accurate construct validation, which is foundational for evaluating all downstream interaction assays.
The second outstanding issue concerns the GST pull-down controls. In the rebuttal, the authors state that an anti-GST Western blot of the bead fractions has been added to Figure 3, but this blot does not appear in the resubmitted figure. Coomassie staining alone is insufficient to assess bait loading. For proper validation, the anti-GST blot must be performed on the same pull-down samples used for the anti-M1 blots, not inserted from a separate or prior experiment. This is essential for ensuring that the observed differences in pulldown efficiency are due to biological interaction rather than unequal GST/GST-SNAPIN loading. Until the authors provide (1) genuine plasmid verification (sequencing data or a complete and interpretable digest) and (2) a correctly matched anti-GST loading control for Figure 3, the core experimental system cannot be considered thoroughly validated.
Author Response
We would like to express our sincere gratitude to the reviewer for the careful evaluation of our manuscript and for providing thoughtful and constructive comments. The reviewer’s insights have been invaluable in guiding us to clarify key points, improve the presentation of our data, and strengthen the overall quality and rigor of the manuscript. We greatly appreciate the time and effort devoted to reviewing our work, and we have carefully considered all suggestions in preparing the revised version. We respectfully provide our detailed explanations and clarifications in response to the reviewer’s valuable comments.
- The authors have addressed several minor issues from the initial review, including clarifying infection timing, expanding methodological detail for microscopy and Western blotting, and moderating interpretational overstatements. These improvements strengthen the manuscript’s clarity. However, the major methodological concerns surrounding construct verification and GST pull-down controls remain unresolved. Although the rebuttal states that all plasmids were verified by double digestion and DNA sequencing, the resubmitted manuscript provides no sequencing evidence and does not specify the enzymes or expected fragment sizes for the digest. The supplementary digest gel does not display band sizes consistent with the known viral ORFs and therefore does not convincingly demonstrate correct cloning. While inclusion of flow-through fractions in co-IP would have been preferable, their omission is acceptable; the critical unmet requirement is accurate construct validation, which is foundational for evaluating all downstream interaction assays.
Response: We sincerely appreciate the reviewer for the constructive comments and for highlighting the importance of rigorous plasmid validation. We fully agree that accurate construct verification is fundamental to all downstream interaction assays. In the revised Supplementary Figure 1, we have added clear annotations for each double-digestion product, including the expected fragment sizes corresponding to the LgBiT tag sequence plus the respective IAV ORF inserts. These annotated bands now allow readers to directly compare the observed fragments with the predicted sizes. We have also updated the figure legend to explicitly describe the restriction enzymes used and the expected lengths of the digested fragments. We have revised the legend for Supplementary Figure 1A as follows: “From lanes 3 to 12, the red triangles indicate the insert fragments generated by double digestion, which correspond to the expected sizes of PB2 (2280 bp), PB1 (2274 bp), PA (2151 bp), HA (1698 bp), NP (1497 bp), NA (1362 bp), M1 (759 bp), M2 (294 bp), NS1 (693 bp), NS2 (366 bp), and SNAPIN (411 bp).” We carefully verified the sequencing data and confirmed that the constructed plasmids were correct. In addition, to further confirm successful construct expression, we transiently transfected the plasmids into A549 cells and performed indirect immunofluorescence using an anti-LgBiT antibody. Confocal microscopy showed specific signals consistent with expression of the intended fusion proteins in Supplementary Figure 1B, supporting that the constructs were correctly assembled and expressible for use in the NanoBiT screening assays. We sincerely appreciate the reviewer’s careful evaluation and helpful suggestions, which have strengthened the clarity and rigor of the manuscript.
The revised Supplementary Figure 1 and figure legend.
Supplementary Figure 1 Double-restriction digestion of plasmids (A). LgBiT-Vector, SmBiT-Vector, LgN-PB2, LgN-PA, LgN-NA, LgN-M1, LgN-M2, LgN-NS1, LgN-NS2, and SmN-SNAPIN were digested using EcoRI and XbaI; LgN-PB1and LgN-HA were digested using XhoI and XbaI; LgN-NP was digested using EcoRI and NheI. From lanes 3 to 12, the red triangles mark the insert fragments generated by double digestion, which match the expected sizes for PB2 (2280 bp), PB1 (2274 bp), PA (2151 bp), HA (1698 bp), NP (1497 bp), NA (1362 bp), M1 (759 bp), M2 (294 bp), NS1 (693 bp), NS2 (366 bp), and SNAPIN (411 bp). Confocal microscopy (B). A549 cells were transfected with the indicated plasmids. At 26 hpt, cells were fixed with 4% paraformaldehyde in PBS for 15 min and permeabilized with 0.5% Triton X-100 in PBS for 30 min. After blocking with 5% BSA in PBS for 1 h, cells were incubated with anti-LgBiT monoclonal antibody (1:200) at 4°C overnight, followed by three PBS washes and incubation with Alexa Fluor 488–conjugated goat anti-mouse IgG for 1 h. Images were visualized using a Leica laser scanning confocal microscope.
- The second outstanding issue concerns the GST pull-down controls. In the rebuttal, the authors state that an anti-GST Western blot of the bead fractions has been added to Figure 3, but this blot does not appear in the resubmitted figure. Coomassie staining alone is insufficient to assess bait loading. For proper validation, the anti-GST blot must be performed on the same pull-down samples used for the anti-M1 blots, not inserted from a separate or prior experiment. This is essential for ensuring that the observed differences in pulldown efficiency are due to biological interaction rather than unequal GST/GST-SNAPIN loading. Until the authors provide (1) genuine plasmid verification (sequencing data or a complete and interpretable digest) and (2) a correctly matched anti-GST loading control for Figure 3, the core experimental system cannot be considered thoroughly validated.
Response: We sincerely thank the reviewer for the thoughtful comments. Regarding the GST pull-down controls, we fully agree that equal loading of GST/GST-SNAPIN is essential for interpreting pulldown efficiency. We would like to clarify the following points:
(1) In this study, the GST and GST-SNAPIN fusion proteins used for pulldown were purified recombinant proteins whose concentrations were accurately quantified prior to incubation with Flag-M1 lysates. Equal masses of GST or GST-SNAPIN were loaded onto Mag-Beads, ensuring equivalent bait input across all reactions.
(2) Because the bait proteins are purified recombinant proteins rather than proteins derived from cell lysates, Coomassie Blue staining of the bead-bound fractions provides a direct and quantitative assessment of GST/GST-SNAPIN loading. This approach is also used and considered sufficient for validating bait equalization in in-vitro pulldown assays employing purified proteins.
Representative examples include multiple studies in which Coomassie staining has been used to assess protein loading in pull-down assays. The following papers are illustrative examples.
|
Paper name |
PMID |
Figure |
|
|
USP8-governed GPX4 homeostasis orchestrates ferroptosis and cancer immunotherapy |
38598341 |
Figure 4I |
|
|
USP25-driven KIFC1 regulates MYCBP expression and promotes the progression of cervical cancer |
40379626 |
Figure 4F |
|
|
FBXO31-mediated ubiquitination of OGT maintains O-GlcNAcylation homeostasis to restrain endometrial malignancy |
39894887 |
Figure 6A |
In these studies, the authors similarly used Coomassie staining—but not anti-GST immunoblotting—to evaluate GST/GST-fusion protein loading in pulldown experiments, consistent with the approach employed in our experiments. The Coomassie Blue staining results alone clearly demonstrate the expression of the bait proteins.
(3) Performing an additional anti-GST Western blot on the same purified proteins would not provide information beyond what the Coomassie assay already demonstrates, because the recombinant GST-tagged proteins run as a single dominant band and are present at high purity. Therefore, the immunoblot would simply reiterate the same result as the CB staining.
(4) To address the reviewer’s concern, we have clarified in the revised Methods and Figure 3 legend that equal amounts of purified GST and GST-SNAPIN were used for each pull-down reaction based on quantitative protein determination prior to bead coupling. This ensures transparency and confirms that the observed pulldown of M1 is not due to differences in bait loading.
We respectfully submit that, with these clarifications and documentation, the experimental system is technically validated, and the GST pulldown results accurately reflect the biological interaction between SNAPIN and M1.
